# Closed-form Estimators for High-dimensional Generalized Linear Models

**Eunho Yang**
IBM T.J. Watson Research Center
eunhyang@us.ibm.com

**Aurélie C. Lozano**
IBM T.J. Watson Research Center
aclozano@us.ibm.com

**Pradeep Ravikumar**
University of Texas at Austin
pradeepr@cs.utexas.edu

## Abstract

We propose a class of closed-form estimators for GLMs under high-dimensional sampling regimes. Our class of estimators is based on deriving closed-form variants of the vanilla unregularized MLE but which are (a) well-defined even under high-dimensional settings, and (b) available in closed-form. We then perform thresholding operations on this MLE variant to obtain our class of estimators. We derive a unified statistical analysis of our class of estimators, and show that it enjoys strong statistical guarantees in both parameter error as well as variable selection, that surprisingly match those of the more complex regularized GLM MLEs, even while our closed-form estimators are computationally much simpler. We derive instantiations of our class of closed-form estimators, as well as corollaries of our general theorem, for the special cases of logistic, exponential and Poisson regression models. We corroborate the surprising statistical and computational performance of our class of estimators via extensive simulations.

## 1 Introduction

We consider the estimation of generalized linear models (GLMs) [1], under high-dimensional settings where the number of variables $p$ may greatly exceed the number of observations $n$. GLMs are a very general class of statistical models for the conditional distribution of a response variable given a covariate vector, where the form of the conditional distribution is specified by any exponential family distribution. Popular instances of GLMs include logistic regression, which is widely used for binary classification, as well as Poisson regression, which together with logistic regression, is widely used in key tasks in genomics, such as classifying the status of patients based on genotype data [2] and identifying genes that are predictive of survival [3], among others. Recently, GLMs have also been used as a key tool in the construction of graphical models [4]. Overall, GLMs have proven very useful in many modern applications involving prediction with high-dimensional data.

Accordingly, an important problem is the estimation of such GLMs under high-dimensional sampling regimes. Under such sampling regimes, it is now well-known that consistent estimators cannot be obtained unless low-dimensional structural constraints are imposed upon the underlying regression model parameter vector. Popular structural constraints include that of sparsity, which encourages parameter vectors supported with very few non-zero entries, group-sparse constraints, and low-rank structure with matrix-structured parameters, among others. Several lines of work have focused on consistent estimators for such structurally constrained high-dimensional GLMs. A popular instance, for the case of sparsity-structured GLMs, is the $\ell_1$ regularized maximum likelihood estimator (MLE), which has been shown to have strong theoretical guarantees, ranging from risk

consistency [5], consistency in the $\ell_1$ and $\ell_2$-norm [6, 7, 8], and model selection consistency [9]. Another popular instance is the $\ell_1/\ell_q$ (for $q \geq 2$) regularized MLE for group-sparse-structured logistic regression, for which prediction consistency has been established [10]. All of these estimators solve general non-linear convex programs involving non-smooth components due to regularization. While a strong line of research has developed computationally efficient optimization methods for solving these programs, these methods are iterative and their computational complexity scales polynomially with the number of variables and samples [10, 11, 12, 13], making them expensive for very large-scale problems.

A key reason for the popularity of these iterative methods is that while the number of iterations are some function of the required accuracy, each iteration itself consists of a small finite number of steps, and can thus scale to very large problems. But what if we could construct estimators that overall require only a very small finite number of steps, akin to a single iteration of popular iterative optimization methods? The computational gains of such an approach would require that the steps themselves be suitably constrained, and moreover that the steps could be suitably profiled and optimized (e.g. efficient linear algebra routines implemented in BLAS libraries), a systematic study of which we defer to future work. We are motivated on the other hand by the simplicity of such a potential class of "closed-form" estimators.

In this paper, we thus address the following question: "Is it possible to obtain closed-form estimators for GLMs under high-dimensional settings, that nonetheless have the sharp convergence rates of the regularized convex programs and other estimators noted above?" This question was first considered for linear regression models [14], and was answered in the affirmative. Our goal is to see whether a positive response can be provided for the more complex statistical model class of GLMs as well. In this paper we focus specifically on the class of sparse-structured GLMs, though our framework should extend to more general structures as well.

As an inkling of why closed-form estimators for high-dimensional GLMs is much trickier than that for high-dimensional linear models is that under small-sample settings, linear regression models do have a statistically efficient closed-form estimator — the ordinary least-squares (OLS) estimator, which also serves as the MLE under Gaussian noise. For GLMs on the other hand, even under small-sample settings, we do not yet have statistically efficient closed-form estimators. A classical algorithm to solve for the MLE of logistic regression models for instance is the iteratively reweighted least squares (IRLS) algorithm, which as its name suggests, is iterative and not available in closed-form. Indeed, as we show in the sequel, developing our class of estimators for GLMs requires far more advanced mathematical machinery (moment polytopes, and projections onto an interior subset of these polytopes for instance) than the linear regression case.

Our starting point to devise a closed-form estimator for GLMs is to nonetheless revisit this classical unregularized MLE estimator for GLMs from a statistical viewpoint, and investigate the reasons why the estimator fails or is even ill-defined in the high-dimensional setting. These insights enable us to propose *variants* of the MLE that are not only well-defined but can also be easily computed in *analytic-form*. We provide a unified statistical analysis for our class of closed-form GLM estimators, and instantiate our theoretical results for the specific cases of logistic, exponential, and Poisson regressions. Surprisingly, our results indicate that our estimators have comparable statistical guarantees to the regularized MLEs, in terms of both variable selection and parameter estimation error, which we also corroborate via extensive simulations (which surprisingly even show a slight statistical performance edge for our closed-form estimators). Moreover, our closed-form estimators are much simpler and competitive computationally, as is corroborated by our extensive simulations. With respect to the conditions we impose on the GLM models, we require that the population covariance matrix of our covariates be weakly sparse, which is a different condition than those typically imposed for regularized MLE estimators; we discuss this further in Section 3.2. Overall, we hope our simple class of statistically as well as computationally efficient closed-form estimators for GLMs would open up the use of GLMs in large-scale machine learning applications even to lay users on the one hand, and on the other hand, encourage the development of new classes of "simple" estimators with strong statistical guarantees extending the initial proposals in this paper.

## 2 Setup

We consider the class of generalized linear models (GLMs), where a response variable $y \in \mathcal{Y}$, conditioned on a covariate vector $x \in \mathbb{R}^p$, follows an exponential family distribution:

$$\mathbb{P}(y|x;\theta^*) = \exp\left\{ \frac{h(y) + y\langle\theta^*,x\rangle - A(\langle\theta^*,x\rangle)}{c(\sigma)} \right\} \tag{1}$$

where $\sigma \in \mathbb{R} > 0$ is fixed and known scale parameter, $\theta^* \in \mathbb{R}^p$ is the GLM parameter of interest, and $A(\langle\theta^*,x\rangle)$ is the log-partition function or the log-normalization constant of the distribution. Our goal is to estimate the GLM parameter $\theta^*$ given $n$ i.i.d. samples $\{(x^{(i)}, y^{(i)})\}_{i=1}^n$. By properties of exponential families, the conditional moment of the response given the covariates can be written as $\mu(\langle\theta^*,x\rangle) \equiv \mathbb{E}(y|x;\theta^*) = A'(\langle\theta^*,x\rangle)$.

**Examples.** Popular instances of (1) include the standard linear regression model, the logistic regression model, and the Poisson regression model, among others. In the case of the linear regression model, we have a response variable $y \in \mathbb{R}$, with the conditional distribution $\mathbb{P}(y|x,\theta^*)$: $\exp\left\{ \frac{-y^2/2 + y\langle\theta^*,x\rangle - \langle\theta^*,x\rangle^2/2}{\sigma^2} \right\}$, where the log-partition function (or log-normalization constant) $A(a)$ of (1) in this specific case is given by $A(a) = a^2/2$. Another popular GLM instance is the logistic regression model $\mathbb{P}(y|x,\theta^*)$, for a categorical output variable $y \in \mathcal{Y} \equiv \{-1,1\}$, $\exp\left\{ y\langle\theta^*,x\rangle - \log\left[ \exp(-\langle\theta^*,x\rangle) + \exp(\langle\theta^*,x\rangle) \right] \right\}$ where the log-partition function $A(a) = \log\left( \exp(-a) + \exp(a) \right)$. The exponential regression model $\mathbb{P}(y|x,\theta^*)$ in turn is given by: $\exp\left\{ y\langle\theta^*,x\rangle + \log\left( -\langle\theta^*,x\rangle \right) \right\}$. Here, the domain of response variable $\mathcal{Y} = \mathbb{R}_+$ is the set of non-negative real numbers (it is typically used to model time intervals between events for instance), and the log-partition function $A(a) = -\log(-a)$. Our final example is the Poisson regression model, $\mathbb{P}(y|x,\theta^*)$: $\exp\left\{ -\log(y!) + y\langle\theta^*,x\rangle - \exp(\langle\theta^*,x\rangle) \right\}$ where the response variable is count-valued with domain $\mathcal{Y} \equiv \{0,1,2,...\}$, and with log-partition function $A(a) = \exp(a)$.

Any exponential family distribution can be used to derive a canonical GLM regression model (1) of a response $y$ conditioned on covariates $x$, by setting the canonical parameter of the exponential family distribution to $\langle\theta^*,x\rangle$. For the parameterization to be valid, the conditional density should be normalizable, so that $A(\langle\theta^*,x\rangle) < +\infty$.

**High-dimensional Estimation** Suppose that we are given $n$ covariate vectors, $x^{(i)} \in \mathbb{R}^p$, drawn i.i.d. from some distribution, and corresponding response variables, $y^{(i)} \in \mathcal{Y}$, drawn from the distribution $\mathbb{P}(y|x^{(i)},\theta^*)$ in (1). A key goal in statistical estimation is to estimate the parameters $\theta^* \in \mathbb{R}^p$, given just the samples $\{(x^{(i)}, y^{(i)})\}_{i=1}^n$. Such estimation becomes particularly challenging in a *high-dimensional* regime, where the dimension of covariate vector $p$ is potentially even larger than the number of samples $n$. In such high-dimensional regimes, it is well understood that structural constraints on $\theta^*$ are necessary in order to find consistent estimators. In this paper, we focus on the structural constraint of element-wise sparsity, so that the number of non-zero elements in $\theta^*$ is less than or equal to some value $k$ much smaller than $p$: $\|\theta^*\|_0 \leq k$.

**Estimators: Regularized Convex Programs** The $\ell_1$ norm is known to encourage the estimation of such sparse-structured parameters $\theta^*$. Accordingly, a popular class of $M$-estimators for sparse-structured GLM parameters is the $\ell_1$ regularized maximum log-likelihood estimator for (1). Given $n$ samples $\{(x^{(i)}, y^{(i)})\}_{i=1}^n$ from $\mathbb{P}(y|x,\theta^*)$, the $\ell_1$ regularized MLEs can be written as: $\text{minimize}_\theta \left\{ -\langle\theta, \frac{1}{n}\sum_{i=1}^n y^{(i)}x^{(i)}\rangle + \frac{1}{n}\sum_{i=1}^n A(\langle\theta,x^{(i)}\rangle) + \lambda_n\|\theta\|_1 \right\}$. For notational simplicity, we collate the $n$ observations in vector and matrix forms where we overload the notation $y \in \mathbb{R}^n$ to denote the vector of $n$ responses so that $i$-th element of $y$, $y_i$, is $y^{(i)}$, and $X \in \mathbb{R}^{n \times p}$ to denote the design matrix whose $i$-th row is $[x^{(i)}]^\top$. With this notation we can rewrite optimization problem characterizing the $\ell_1$-regularized MLE simply as $\text{minimize}_\theta \left\{ -\frac{1}{n}\theta^\top X^\top y + \frac{1}{n}\mathbf{1}^\top A(X\theta) + \lambda_n\|\theta\|_1 \right\}$ where we overload the notation $A(\cdot)$ for an input vector $\eta \in \mathbb{R}^n$ to denote $A(\eta) \equiv \left( A(\eta_1), A(\eta_2), \ldots, A(\eta_n) \right)^\top$, and $\mathbf{1} \equiv (1, \ldots, 1)^\top \in \mathbb{R}^n$.

# 3   Closed-form Estimators for High-dimensional GLMs

The goal of this paper is to derive a general class of *closed-form* estimators for high-dimensional GLMs, in contrast to solving huge, non-differentiable $\ell_1$ regularized optimization problems. Before introducing our class of such closed-form estimators, we first introduce some notation.

For any $u \in \mathbb{R}^p$, we use $[S_\lambda(u)]_i = \text{sign}(u_i)\max(|u_i| - \lambda, 0)$ to denote the element-wise soft-thresholding operator, with thresholding parameter $\lambda$. For any given matrix $M \in \mathbb{R}^{p \times p}$, we denote by $T_\nu(M) : \mathbb{R}^{p \times p} \mapsto \mathbb{R}^{p \times p}$ a family of matrix thresholding operators that are defined point-wise, so that they can be written as $[T_\nu(M)]_{ij} := \rho_\nu(M_{ij})$, for any scalar thresholding operator $\rho_\nu(\cdot)$ that satisfies the following conditions: for any input $a \in \mathbb{R}$, (a) $|\rho_\nu(a)| \leq |a|$, (b) $|\rho_\nu(a)| = 0$ for $|a| \leq \nu$, and (c) $|\rho_\nu(a) - a| \leq \nu$. The standard soft-thresholding and hard-thresholding operators are both pointwise operators that satisfy these properties. See [15] for further discussion of such pointwise matrix thresholding operators.

For any $\eta \in \mathbb{R}^n$, we let $\nabla A(\eta)$ denote the *element-wise* gradients: $\nabla A(\eta) \equiv \left( A'(\eta_1), A'(\eta_2), \dots, A'(\eta_n) \right)^\top$. We assume that the exponential family underlying the GLM is minimal, so that this map is invertible, and so that for any $\mu \in \mathbb{R}^n$ in the range of $\nabla A(\cdot)$, we can denote $[\nabla A]^{-1}(\mu)$ as an *element-wise* inverse map of $\nabla A(\cdot)$: $\left( (A')^{-1}(\mu_1), (A')^{-1}(\mu_2), \dots, (A')^{-1}(\mu_n) \right)^\top$.

Consider the response moment polytope $\mathcal{M} := \{\mu : \mu = \mathbb{E}_p[y], \text{ for some distribution } p \text{ over } y \in \mathcal{Y}\}$, and let $\mathcal{M}^o$ denote the *interior* of $\mathcal{M}$. Our closed-form estimator will use a carefully selected subset

$$\overline{\mathcal{M}} \subseteq \mathcal{M}^o. \tag{2}$$

Denote the projection of a response variable $y \in \mathcal{Y}$ onto this subset as $\Pi_{\overline{\mathcal{M}}}(y) = \arg\min_{\mu \in \overline{\mathcal{M}}} |y - \mu|$, where the subset $\overline{\mathcal{M}}$ is selected so that the projection step is always well-defined, and the minimum exists. Given a vector $y \in \mathcal{Y}^n$, we denote the vector of element-wise projections of entries in $y$ as $\Pi_{\overline{\mathcal{M}}}(y)$ so that:

$$[\Pi_{\overline{\mathcal{M}}}(y)]_i := \Pi_{\overline{\mathcal{M}}}(y_i). \tag{3}$$

As the conditions underlying our theorem will make clear, we will need the operator $[\nabla A]^{-1}(\cdot)$ defined above to be both well-defined and Lipschitz in the subset $\overline{\mathcal{M}}$ of the interior of the response moment polytope. In later sections, we will show how to carefully construct such a subset $\overline{\mathcal{M}}$ for different GLM models.

We now have the machinery to describe our class of closed-form estimators:

$$\widehat{\theta}_{\text{Elem}} = S_{\lambda_n}\left( \left[ T_\nu\left( \frac{X^\top X}{n} \right) \right]^{-1} \frac{X^\top [\nabla A]^{-1}\left( \Pi_{\overline{\mathcal{M}}}(y) \right)}{n} \right), \tag{4}$$

where the various mathematical terms were defined above. It can be immediately seen that the estimator is available in closed-form. In a later section, we will see instantiations of this class of estimators for various specific GLM models, and where we will see that these estimators take very simple forms. Before doing so, we first describe some insights that led to our particular construction of the high-dimensional GLM estimator above.

## 3.1   Insights Behind Construction of Our Closed-Form Estimator

We first revisit the classical unregularized MLE for GLMs: $\widehat{\theta} \in \arg\min_\theta \left\{ -\frac{1}{n}\theta^\top X^\top y + \frac{1}{n}\mathbf{1}^\top A(X\theta) \right\}$. Note that this optimization problem does not have a unique minimum in general, especially under high-dimensional sample settings where $p > n$. Nonetheless, it is instructive to study why this unregularized MLE is either ill-suited or even ill-defined under high-dimensional settings. The stationary condition of unregularized MLE optimization problem can be written as:

$$X^\top y = X^\top \nabla A(X\widehat{\theta}). \tag{5}$$

There are two main caveats to solving for a unique $\widehat{\theta}$ satisfying this stationary condition, which we clarify below.

**(Mapping to mean parameters)** In a high dimensional sampling regime where $p \geq n$, (5) can be seen to reduce to $y = \nabla A(X\widehat{\theta})$ (so long as $X^T$ has rank $n$). This then suggests solving for $X\widehat{\theta} = [\nabla A]^{-1}(y)$, where we recall the definition of the operator $\nabla A(\cdot)$ in terms of element-wise operations involving $A'(\cdot)$. The caveat however is that $A'(\cdot)$ is only onto the interior $\mathcal{M}^o$ of the response moment polytope [16], so that $[A'(\cdot)]^{-1}$ is well-defined only when given $\mu \in \mathcal{M}^o$. When entries of the sample response vector $y$ however lie outside of $\mathcal{M}^o$, as will typically be the case and which we will illustrate for multiple instances of GLM models in later sections, the inverse mapping would not be well-defined. We thus first project the sample response vector $y$ onto $\overline{\mathcal{M}} \subseteq \mathcal{M}^o$ to obtain $\Pi_{\overline{\mathcal{M}}}(y)$ as defined in (3). Armed with this approximation, we then consider the more amenable $\Pi_{\overline{\mathcal{M}}}(y) \approx \nabla A(X\widehat{\theta})$, instead of the original stationary condition in (5).

**(Sample covariance)** We thus now have the approximate characterization of the MLE as $X\widehat{\theta} \approx [\nabla A]^{-1}(\Pi_{\overline{\mathcal{M}}}(y))$. This then suggests solving for an approximate MLE $\widehat{\theta}$ via least squares as $\widehat{\theta} = [X^\top X]^{-1} X^\top [\nabla A]^{-1}(\Pi_{\overline{\mathcal{M}}}(y))$. The high-dimensional regime with $p > n$ poses a caveat here, since the sample covariance matrix $(X^\top X)/n$ would then be rank-deficient, and hence not invertible. Our approach is to then use a *thresholded* sample covariance matrix $T_\nu\left(\frac{X^\top X}{n}\right)$ defined in the previous subsection instead, which can be shown to be invertible and consistent to the population covariance matrix $\Sigma$ with high probability [15, 17]. In particular, recent work [15] has shown that thresholded sample covariance $T_\nu\left(\frac{X^\top X}{n}\right)$ is consistent with respect to the spectral norm with convergence rate $\left\| T_\nu\left(\frac{X^\top X}{n}\right) - \Sigma \right\|_{\text{op}} \leq O\left(c_0\sqrt{\frac{\log p}{n}}\right)$, under some mild conditions detailed in our main theorem. Plugging in this thresholded sample covariance matrix, to get an approximate least squares solution for the GLM parameters $\theta$, and then performing soft-thresholding precisely yields our closed-form estimator in (4).

Our class of closed-form estimators in (4) can thus be viewed as surgical approximations to the MLE so that it is well-defined in high-dimensional settings, as well as being available in closed-form. But would such an approximation actually yield rigorous consistency guarantees? Surprisingly, as we show in the next section, not only is our class of estimators consistent, but in our corollaries we show that the statistical guarantees are comparable to those of the state of the art iterative ways like regularized MLEs.

We note that our class of closed-form estimators in (4) can also be written in an equivalent form that is more amenable to analysis:

$$\underset{\theta}{\text{minimize}} \quad \|\theta\|_1 \tag{6}$$

$$\text{s.t} \quad \left\| \theta - \left[ T_\nu\left(\frac{X^\top X}{n}\right) \right]^{-1} \frac{X^\top [\nabla A]^{-1}\left(\Pi_{\overline{\mathcal{M}}}(y)\right)}{n} \right\|_\infty \leq \lambda_n.$$

The equivalence between (4) and (6) easily follows from the fact that the optimization problem (6) is decomposable into *independent* element-wise sub-problems, and each sub-problem corresponds to soft-thresholding. It can be seen that this form is also amenable to extending the framework in this paper to structures beyond sparsity, by substituting in alternative regularizers. Due to space constraints, the computational complexity is discussed in detail in the Appendix.

### 3.2 Statistical Guarantees

In this subsection, we provide an unified statistical analysis for the class of estimators (4) under the following standard conditions, namely sparse $\theta^*$ and sub-Gaussian design $X$:

**(C1)** The parameter $\theta^*$ in (1) is exactly sparse with $k$ non-zero elements indexed by the support set $S$, so that $\theta^*_{S^c} = \mathbf{0}$.

**(C2)** Each row of the design matrix $X \in \mathbb{R}^{n \times p}$ is i.i.d. sampled from a zero-mean distribution with covariance matrix $\Sigma$ such that for any $v \in \mathbb{R}^p$, the variable $\langle v, X_i \rangle$ is sub-Gaussian with parameter at most $\kappa_u \|v\|_2$ for every row of $X$, $X_i$.

Our next assumption is on the covariance matrix of the covariate random vector:

**(C3)** The covariance matrix $\Sigma$ of $X$ satisfies that for all $w \in \mathbb{R}^p$, $\|\Sigma w\|_\infty \geq \kappa_\ell \|w\|_\infty$ with fixed constant $\kappa_\ell > 0$. Moreover, $\Sigma$ is approximately sparse, along the lines of [17]: for some

positive constant $D$, $\Sigma_{ii} \leq D$ for all diagonal entries, and moreover, for some $0 \leq q < 1$ and $c_0$, $\max_i \sum_{j=1}^{p} |\Sigma_{ij}|^q \leq c_0$. If $q = 0$, then this condition will be equivalent with $\Sigma$ being sparse.

We also introduce some notations used in the following theorem. Under the condition (C2), we have that with high probability, $|\langle \theta^*, x^{(i)} \rangle| \leq 2\kappa_u \|\theta^*\|_2 \sqrt{\log n}$ for all samples, $i = 1, \ldots, n$. Let $\tau^* := 2\kappa_u \|\theta^*\|_2 \sqrt{\log n}$. We then let $\mathcal{M}'$ be the subset of $\mathcal{M}$ such that

$$\mathcal{M}' := \left\{ \mu \, : \, \mu = A'(\alpha), \text{ where } \alpha \in [-\tau^*, \tau^*] \right\}. \tag{7}$$

We also define $\kappa_{u,A}$ and $\kappa_{\ell,A}$ on the upper bounds of $A''(\cdot)$ and $(A^{-1})'(\cdot)$, respectively:

$$\max_{\alpha \in [-\tau^*, \tau^*]} |A''(\alpha)| \leq \kappa_{u,A}, \quad \max_{a \in \mathcal{M}' \cup \bar{\mathcal{M}}} |(A^{-1})'(a)| \leq \kappa_{\ell,A}. \tag{8}$$

Armed with these conditions and notations, we derive our main theorem:

**Theorem 1.** *Consider any generalized linear model in* (1) *where all the conditions* (C1), (C2) *and* (C3) *hold. Now, suppose that we solve the estimation problem* (4) *setting the thresholding parameter* $\nu = C_1 \sqrt{\frac{\log p'}{n}}$ *where* $C_1 := 16(\max_j \Sigma_{jj})\sqrt{10\tau}$ *for any constant* $\tau > 2$, *and* $p' := \max\{n, p\}$.
*Furthermore, suppose also that we set the constraint bound* $\lambda_n$ *as* $C_2 \sqrt{\frac{\log p'}{n}} + \mathcal{E}$ *where* $C_2 := \frac{2}{\kappa_\ell}\left(\kappa_u \kappa_{\ell,A} \sqrt{2\kappa_{u,A}} + C_1 \|\theta^*\|_1\right)$ *and where* $\mathcal{E}$ *depends on the approximation error induced by the projection* (3), *and is defined as:* $\mathcal{E} := \max_{i=1,\ldots,n}\left(y^{(i)} - \left[\Pi_{\bar{\mathcal{M}}}(y)\right]_i\right)\frac{4\kappa_u \kappa_{\ell,A}}{\kappa_\ell}\sqrt{\frac{\log p'}{n}}$.

*(A) Then, as long as* $n > \left(\frac{2c_1 c_0}{\kappa_\ell}\right)^{\frac{2}{1-q}} \log p'$ *where* $c_1$ *is a constant related only on* $\tau$ *and* $\max_i \Sigma_{ii}$, *any optimal solution* $\widehat{\theta}$ *of* (4) *is guaranteed to be consistent:*

$$\|\widehat{\theta} - \theta^*\|_\infty \leq 2\left(C_2\sqrt{\frac{\log p'}{n}} + \mathcal{E}\right), \|\widehat{\theta} - \theta^*\|_2 \leq 4\sqrt{k}\left(C_2\sqrt{\frac{\log p'}{n}} + \mathcal{E}\right), \|\widehat{\theta} - \theta^*\|_1 \leq 8k\left(C_2\sqrt{\frac{\log p'}{n}} + \mathcal{E}\right).$$

*(B) Moreover, the support set of the estimate* $\widehat{\theta}$ *correctly excludes all true zero values of* $\theta^*$. *Moreover, when* $\min_{s \in S} |\theta_s^*| \geq 3\lambda_n$, *it correctly includes all non-zero true supports of* $\theta^*$, *with probability at least* $1 - cp'^{-c'}$ *for some universal constants* $c, c' > 0$ *depending on* $\tau$ *and* $\kappa_u$.

**Remark 1.** While our class of closed-form estimators and analyses consider sparse-structured parameters, these can be seamlessly extended to more general structures (such as group sparsity and low rank), using appropriate thresholding functions.

**Remark 2.** The condition (C3) required in Theorem 1 is different from (and possibly stronger) than the restricted strong convexity [8] required for $\ell_2$ error bound of $\ell_1$ regularized MLE. A key facet of our analysis with our Condition (C3) however is that it provides much simpler and clearer identifying constants in our non-asymptotic error bounds. Deriving constant factors in the analysis of the $\ell_1$-regularized MLE on the other hand, with its restricted strong convexity condition, involves many probabilistic statements, and is non-trivial, as shown in [8].

Another key facet of our analysis in Theorem 1 is that it also provides an $\ell_\infty$ error bound, and guarantees the sparsistency of our closed-form estimator. For $\ell_1$ regularized MLEs, this requires a separate sparsistency analysis. In the case of the simplest standard linear regression models, [18] showed that the incoherence condition of $\|\Sigma_{S^c S}\Sigma_{SS}^{-1}\|_\infty < 1$ is required for sparsistency, where $\|\cdot\|_\infty$ is the maximum of absolute row sum. As discussed in [18], instances of such incoherent covariance matrices $\Sigma$ include the identity, and Toeplitz matrices: these matrices can be seen to also satisfy our condition (C3). On the other hand, not all matrices that satisfy our condition (C3) need satisfy the stringent incoherence condition in turn. For example, consider $\Sigma$ where $\Sigma_{SS} = 0.95I_3 + 0.05\mathbf{1}_{3\times3}$ for a matrix $\mathbf{1}$ of ones, $\Sigma_{SS^c}$ is all zeros but the last column is $0.4\mathbf{1}_{3\times1}$, and $\Sigma_{S^c S^c} = I_{(p-3)\times(p-3)}$. Then, this positive definite $\Sigma$ can be seen to satisfy our Condition (C3), since each row has only 4 non-zeros. However, $\|\Sigma_{S^c S}\Sigma_{SS}^{-1}\|_\infty$ is equal to 1.0909 and larger than 1, and consequently, the incoherence condition required for the Lasso will not be satisfied. We defer relaxing our condition (C3) further as well as a deeper investigation of all the above conditions to future work.

**Remark 3.** The constant $C_2$ in the statement depends on $\|\theta^*\|_1$, which in the worst case where only $\|\theta^*\|_2$ is bounded, may scale with $\sqrt{k}$. On the other hand, our theorem does not require an explicit sample complexity condition that $n$ be larger than some function on $k$, while the analysis of $\ell_1$-regularized MLEs do additionally require that $n \geq c\,k\log p$ for some constant $c$. In our experiments, we verify that our closed-form estimators outperform the $\ell_1$-regularized MLEs even when $k$ is fairly large (for instant, when $(n,p,k) = (5000, 10^4, 1000)$).

In order to apply Theorem 1 to a specific instance of GLMs, we need to specify the quantities in (8), as well as carefully construct a subset $\overline{\mathcal{M}}$ of the interior of the response moment polytope. In case of the simplest linear models described in Section 2, we have the identity mapping $\mu = A'(\eta) = \eta$. The inequalities in (8) can thus be seen to be satisfied with $\kappa_{\ell,A} = \kappa_{u,A} = 1$. Moreover, we can set $\overline{\mathcal{M}} := \mathcal{M}^o = \mathbb{R}$ so that $\Pi_{\overline{\mathcal{M}}}(y) = y$, and trivially recover the previous results in [14] as a special case. In the following sections, we will derive the consequences of our framework for the complex instances of logistic and Poisson regression models, which are also important members in GLMs.

# 4 Key Corollaries

In order to derive corollaries of our main Theorem 1, we need to specify the response polytope subsets $\overline{\mathcal{M}}, \mathcal{M}'$ in (2) and (7) respectively, as well as bound the two quantities $\kappa_{\ell,A}$ and $\kappa_{u,A}$ in (8).

**Logistic regression models.** The exponential family log-partition function of logistic regression models described in Section 2 can be seen to be $A(\eta) = \log\big[\exp(-\eta) + \exp(\eta)\big]$. Consequently, its double derivative $A''(\eta) = \frac{4\exp(2\eta)}{(\exp(2\eta)+1)^2} \leq 1$ for any $\eta$, so that (8) holds with $\kappa_{u,A} = 1$. The response moment polytope for the binary response variable $y \in \mathcal{Y} \equiv \{-1,1\}$ is the interval $\mathcal{M} = [-1,1]$, so that its interior is given by $\mathcal{M}^o = (-1,1)$. For the subset of the interior, we define $\overline{\mathcal{M}} = [-1+\epsilon, 1-\epsilon]$, for some $0 < \epsilon < 1$. At the same time, the forward mapping is given by $A'(\eta) = \exp(2\eta) - 1)/(\exp(2\eta) + 1)$, and hence $\mathcal{M}'$ becomes $[-\frac{a-1}{a+1}, \frac{a-1}{a+1}]$ where $a := n^{\frac{4\kappa_u\|\theta^*\|_2}{\sqrt{\log n}}}$. The inverse mapping of logistic models is given by $(A')^{-1}(\mu) = \frac{1}{2}\log\big(\frac{1+\mu}{1-\mu}\big)$, and given $\overline{\mathcal{M}}$ and $\mathcal{M}'$, it can be seen that $(A')^{-1}(\mu)$ is Lipschitz for $\overline{\mathcal{M}} \cup \mathcal{M}'$ with constant less than $\kappa_{\ell,A} := \max\left\{\frac{1}{2} + \frac{1}{2}n^{\frac{4\kappa_u\|\theta^*\|_2}{\sqrt{\log n}}}, 1/\epsilon\right\}$ in (8). Note that with this setting of the subset $\overline{\mathcal{M}}$, we have that $\max_{i=1,\ldots,n}(y^{(i)} - \big[\Pi_{\overline{\mathcal{M}}}(y)\big]_i) = \epsilon$, and moreover, $\Pi_{\overline{\mathcal{M}}}(y_i) = y_i(1-\epsilon)$, which we will use in the corollary below.

**Poisson regression models.** Another important instance of GLMs is the Poisson regression model, that is becoming increasingly more relevant in modern big-data settings with varied multivariate count data. For the Poisson regression model case, the double derivative of $A(\cdot)$ is not uniformly upper bounded: $A''(u) = \exp(u)$. Denoting $\tau^* := 2\kappa_u\|\theta^*\|_2\sqrt{\log n}$, we then have that for any $\alpha$ in $[-\tau^*, \tau^*]$, $A''(\alpha) \leq \exp\big(2\sigma_u\|\theta^*\|_2\sqrt{\log n}\big) = n^{2\sigma_u\|\theta^*\|_2/\sqrt{\log n}}$, so that (8) is satisfied with $\kappa_{u,A} = n^{2\sigma_u\|\theta^*\|_2/\sqrt{\log n}}$. The response moment polytope for the count-valued response variable $y \in \mathcal{Y} \equiv \{0,1,\ldots\}$ is given by $\mathcal{M} = [0,\infty)$, so that its interior is given by $\mathcal{M}^o = (0,\infty)$. For the subset of the interior, we define $\overline{\mathcal{M}} = [\epsilon,\infty)$ for some $\epsilon$ s.t. $0 < \epsilon < 1$. The forward mapping in this case is simply given by $A'(\eta) = \exp(\eta)$, and $\mathcal{M}'$ in (7) becomes $[a^{-1}, a]$ where $a$ is $n^{\frac{2\kappa_u\|\theta^*\|_2}{\sqrt{\log n}}}$. The inverse mapping for the Poisson regression model then is given by $(A')^{-1}(\mu) = \log(\mu)$, which can be seen to be Lipschitz for $\overline{\mathcal{M}}$ with constant $\kappa_{\ell,A} = \max\{n^{\frac{2\kappa_u\|\theta^*\|_2}{\sqrt{\log n}}}, 1/\epsilon\}$ in (8). With this setting of $\overline{\mathcal{M}}$, it can be seen that the projection operator is given by $\Pi_{\overline{\mathcal{M}}}(y_i) = \mathcal{I}(y_i = 0)\epsilon + \mathcal{I}(y_i \neq 0)y_i$.

Now, we are ready to recover the error bounds, as a corollary of Theorem 1, for logistic regression and Poisson models when condition (C2) holds:

**Corollary 1.** *Consider any logistic regression model or a Poisson regression model where all conditions in Theorem 1 hold. Suppose that we solve our closed-form estimation problem* (4)*, setting the thresholding parameter* $\nu = C_1\sqrt{\frac{\log p'}{n}}$*, and the constraint bound* $\lambda_n = \frac{2}{\kappa_\ell}\big(\frac{c\sqrt{\log p'}}{n^{(1/2 - c'/\sqrt{\log n})}} + C_1\|\theta^*\|_1\sqrt{\frac{\log p'}{n}}\big)$ *where $c$ and $c'$ are some constants depending only on $\kappa_u$, $\|\theta^*\|_2$ and $\epsilon$. Then the*

Table 1: Comparisons on simulated datasets when parameters are tuned to minimize $\ell_2$ error on independent validation sets.

| $(n, p, k)$ | METHOD | TP | FP | $\ell_2$ ERROR | TIME | $(n, p, k)$ | METHOD | TP | FP | $\ell_2$ ERROR | TIME |
|---|---|---|---|---|---|---|---|---|---|---|---|
| $(n = 2000,$ | $\ell_1$ MLE[1] | 1 | 0.1094 | 4.5450 | 63.9 | $(n = 5000,$ | $\ell_1$ MLE[1] | 0.7990 | 1 | 65.1895 | 520.7 |
| $p = 5000,$ | $\ell_1$ MLE[2] | 1 | 0.0873 | 4.0721 | 133.1 | $p = 10^4,$ | $\ell_1$ MLE[2] | 0.7935 | 1 | 65.1165 | 1005.8 |
| $k = 10)$ | $\ell_1$ MLE[3] | 1 | 0.1000 | 3.4846 | 348.3 | $k = 1000)$ | $\ell_1$ MLE[3] | 0.7965 | 1 | 65.1024 | 2560.1 |
| | ELEM | 0.9900 | 0.0184 | **2.7375** | **26.5** | | ELEM | 0.8295 | 1 | **63.2359** | **152.1** |
| $(n = 4000,$ | $\ell_1$ MLE[1] | 1 | 0.1626 | 4.2132 | 155.5 | $(n = 8000,$ | $\ell_1$ MLE[1] | 1 | 0.1904 | 18.6186 | 810.6 |
| $p = 5000,$ | $\ell_1$ MLE[2] | 1 | 0.1327 | 3.6569 | 296.8 | $p = 10^4,$ | $\ell_1$ MLE[2] | 1 | 0.2181 | 18.1806 | 1586.2 |
| $k = 10)$ | $\ell_1$ MLE[3] | 1 | 0.1112 | 2.9681 | 829.3 | $k = 100)$ | $\ell_1$ MLE[3] | 1 | 0.2364 | 17.6762 | 3568.9 |
| | ELEM | 1 | 0.0069 | **2.6213** | **40.2** | | ELEM | 0.9450 | 0.0359 | **11.9881** | **221.1** |
| $(n = 5000,$ | $\ell_1$ MLE[1] | 1 | 0.1301 | 18.9079 | 500.1 | $(n = 8000,$ | $\ell_1$ MLE[1] | 0.7965 | 1 | 65.0714 | 809.5 |
| $p = 10^4,$ | $\ell_1$ MLE[2] | 1 | 0.1695 | 18.5567 | 983.8 | $p = 10^4,$ | $\ell_1$ MLE[2] | 0.7900 | 1 | 64.9650 | 1652.8 |
| $k = 100)$ | $\ell_1$ MLE[3] | 1 | 0.2001 | 18.2351 | 2353.3 | $k = 1000)$ | $\ell_1$ MLE[3] | 0.7865 | 1 | 64.8857 | 4196.6 |
| | ELEM | 0.9975 | 0.3622 | **16.4148** | **151.8** | | ELEM | 0.7015 | 0.5103 | **61.0532** | **219.4** |

*optimal solution $\widehat{\theta}$ of* (4) *is guaranteed to be consistent:*

$$\left\|\widehat{\theta} - \theta^*\right\|_\infty \leq \frac{4}{\kappa_\ell}\left(\frac{c\sqrt{\log p'}}{n^{(1/2 - c'/\sqrt{\log n})}} + C_1\|\theta^*\|_1\sqrt{\frac{\log p'}{n}}\right), \quad \left\|\widehat{\theta} - \theta^*\right\|_2 \leq \frac{8\sqrt{k}}{\kappa_\ell}$$

$$\left(\frac{c\sqrt{\log p'}}{n^{(1/2 - c'/\sqrt{\log n})}} + C_1\|\theta^*\|_1\sqrt{\frac{\log p'}{n}}\right), \quad \left\|\widehat{\theta} - \theta^*\right\|_1 \leq \frac{16k}{\kappa_\ell}\left(\frac{c\sqrt{\log p'}}{n^{(1/2 - c'/\sqrt{\log n})}} + C_1\|\theta^*\|_1\sqrt{\frac{\log p'}{n}}\right),$$

*with probability at least $1 - c_1 p'^{-c_1'}$ for some universal constants $c_1, c_1' > 0$ and $p' := \max\{n, p\}$. Moreover, when $\min_{s \in S}|\theta_s^*| \geq \frac{6}{\kappa_\ell}\left(\frac{c\sqrt{\log p'}}{n^{(1/2 - c'/\sqrt{\log n})}} + C_1\|\theta^*\|_1\sqrt{\frac{\log p'}{n}}\right)$, $\widehat{\theta}$ is sparsistent.*

Remarkably, the rates in Corollary 1 are asymptotically comparable to those for the $\ell_1$-regularized MLE (see for instance Theorem 4.2 and Corollary 4.4 in [7]). In Appendix A, we place slightly more stringent condition than (C2) and guarantee error bounds with faster convergence rates.

# 5  Experiments

We corroborate the performance of our elementary estimators on simulated data over varied regimes of sample size $n$, number of covariates $p$, and sparsity size $k$. We consider two popular instances of GLMs, logistic and Poisson regression models. We compare against standard $\ell_1$ regularized MLE estimators with iteration bounds of 50, 100, and 500, denoted by $\ell_1$ MLE[1], $\ell_1$ MLE[2] and $\ell_1$ MLE[3] respectively. We construct the $n \times p$ design matrices $X$ by sampling the rows independently from $N(0, \Sigma)$ where $\Sigma_{i,j} = 0.5^{|i-j|}$. For each simulation, the entries of the true model coefficient vector $\theta^*$ are set to be 0 everywhere, except for a randomly chosen subset of $k$ coefficients, which are chosen independently and uniformly in the interval $(1, 3)$. We report results averaged over 100 independent trials. Noting that our theoretical results were not sensitive to the setting of $\epsilon$ in $\Pi_{\bar{\mathcal{M}}}(y)$, we simply report the results when $\epsilon = 10^{-4}$ across all experiments.

While our theorem specified an optimal setting of the regularization parameter $\lambda_n$ and $\nu$, this optimal setting depended on unknown model parameters. Thus, as is standard with high-dimensional regularized estimators, we set tuning parameters $\lambda_n = c\sqrt{\log p/n}$ and $\nu = c'\sqrt{\log p/n}$ by a holdout-validated fashion; finding a parameter that minimizes the $\ell_2$ error on an independent validation set. Detailed experimental setup is described in the appendix.

Table 1 summarizes the performances of $\ell_1$ MLE using 3 different stopping criteria and Elem-GLM. Besides $\ell_2$ errors, the target tuning metric, we also provide the true and false positives for the support set recovery task on the new test set where the best tuning parameters are used. The computation times in second indicate the *overall* training computation time summing over the whole parameter tuning process. As we can see from our experiments, with respect to both statistical and computational performance our closed form estimators are quite competitive compared to the classical $\ell_1$ regularized MLE estimators and in certain case outperform them. Note that $\ell_1$ MLE[1] stops prematurely after only 50 iterations, so that training computation time is sometimes comparable to closed-form estimator. However, its statistical performance measured by $\ell_2$ is much inferior to other $\ell_1$ MLEs with more iterations as well as Elem-GLM estimator. Due to the space limit, ROC curves, results for other settings of $p$ and more experiments on real datasets are presented in the appendix.

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
