[Supplementary Material]

# Appendix

## A  Statistical Analysis: When Covariates Are Bounded

In this section, we consider our closed-form estimators in the case when covariates $x$ are bounded:

**(C2')** For every $i = 1, \ldots, n$, $\|x^{(i)}\|_2$ is bounded, so that $\langle \theta^*, x^{(i)} \rangle$ is also bounded by $\eta_{\max}$,

under which the behavior of $\ell_1$ regularized MLEs is well studied in several works [6, 8]. The case where the covariates $x$ are strictly bounded requires a more straightforward analysis. In particular, we can provide simplified specifications of the bounds $\kappa_{\ell,A}$ and $\kappa_{u,A}$ in (8), as well as of the response polytope subsets $\mathcal{M}'$ in (7).

Specifically, suppose that $|\langle \theta^*, x^{(i)} \rangle| \leq \eta_{\max}$ for all $i = 1, \ldots, n$. We then define the following response polytope subset $\mathcal{M}'$ such that

$$\mathcal{M}' := \left\{ \mu \,:\, \mu = A'(\alpha) \,,\text{ where } \alpha \in [-\eta_{\max}, \eta_{\max}] \right\}. \tag{9}$$

The bound $\kappa_{\ell,A}$ in (8) should hold as before but with this simpler subset $\mathcal{M}'$, so that $\max_{a \in \mathcal{M}' \cup \bar{\mathcal{M}}} |[A^{-1}]'(a)| \leq \kappa_{\ell,A}$. In addition, we modify the inequality for the bound $\kappa_{u,A}$ in (8) so that $\max_{\alpha \in [-\eta_{\max}, \eta_{\max}]} |A''(\alpha)| \leq \kappa_{u,A}$.

**Logistic regression models.**   Since the double derivative of the log partition function in this case is uniformly bounded, (8) still holds with $\kappa_{u,A} = 1$. On the other hand, when all covariates are bounded and hence $\langle \theta^*, x^{(i)} \rangle$ is bounded by $\eta_{\max}$, it can be shown that (8) for a new $\mathcal{M}'$ in (9) holds when $\kappa_{\ell,A} = \max\{\frac{1}{2} + \frac{1}{2}\exp(2\eta_{\max}), 1/\epsilon\}$.

**Poisson regression models.**   Similarly in the logistics case, if $\langle \theta^*, x^{(i)} \rangle$ is bounded by $\eta_{\max}$, then $\kappa_{\ell,A} = \max\{\exp(\eta_{\max}), 1/\epsilon\}$ and $\kappa_{u,A} = \exp(\eta_{\max})$ from the definitions of $A''(\cdot)$ and $[A^{-1}](\cdot)$ of Poisson models. Another example is the exponential regression case, which is provided in the appendix due to the space limit.

Given $\kappa_{\ell,A}$, $\kappa_{u,A}$ and $\bar{\mathcal{M}}$ as specified above, we can recover the following error bounds as a corollary of Theorem 1 when covariates are bounded:

**Corollary 2.** *Consider any logistic regression model or a Poisson regression model where conditions* (C1)*,* (C3) *as well as* (C2') *hold. Suppose that we solve our closed-form estimation problem* (4)*, setting the thresholding parameter* $\nu = C_1 \sqrt{\frac{\log p'}{n}}$*, and the constraint bound* $\lambda_n = \frac{2}{\kappa_\ell}(C_1 + C_1')\sqrt{\frac{\log p'}{n}}$ *where* $C_1'$ *is some constant depending only on* $\eta_{\max}$ *and* $\epsilon$*. Then the optimal solution* $\widehat{\theta}$ *of* (4) *is guaranteed to be consistent:*

$$\|\widehat{\theta} - \theta^*\|_\infty \leq \frac{4}{\kappa_\ell}(C_1 + C_1')\sqrt{\frac{\log p'}{n}},$$

$$\|\widehat{\theta} - \theta^*\|_2 \leq \frac{8}{\kappa_\ell}(C_1 + C_1')\sqrt{\frac{k \log p'}{n}},$$

$$\|\widehat{\theta} - \theta^*\|_1 \leq \frac{16k}{\kappa_\ell}(C_1 + C_1')\sqrt{\frac{\log p'}{n}},$$

*with probability at least* $1 - c_1 p'^{-c_1'}$ *for some universal constants* $c_1, c_1' > 0$ *and* $p' := \max\{n, p\}$*. Moreover, when* $\min_{s \in S} |\theta_s^*| \geq \frac{6}{\kappa_\ell}(C_1 + C_1')\sqrt{\frac{\log p'}{n}}$*,* $\widehat{\theta}$ *is sparsistent.*

We remark that the rates in Corollary 2 are asymptotically the same as those for standard $\ell_1$-regularized MLE estimators (see for instance Theorem 5 in [6]). This is especially remarkable given the simplicity of our framework.

## B  Computational Complexity

Computing our closed-form estimator in (6) requires solving a linear system (and not necessarily a matrix inversion). In general, the time complexity of solving a linear system depends on the

sparsity of the matrix. For instance, conjugate gradient as a direct method has a time complexity of $O(ps)$ where $s$ is the number of non-zero entries of $T_\nu\left(\frac{X^\top X}{n}\right)$ [19], while iterative conjugate gradient solvers have a time complexity of $O(s\sqrt{\kappa_{\text{cone}}})$, where $\kappa_{\text{cone}}$ is the condition number of $T_\nu\left(\frac{X^\top X}{n}\right)$ [19]. Recently developed solvers considerably improve upon these time complexities by further exploiting the structure of of $T_\nu\left(\frac{X^\top X}{n}\right)$. For instance, for the case where $T_\nu\left(\frac{X^\top X}{n}\right)$ is diagonally dominant, the linear system can be solved in time $O(s\log p^{1/2})$ using the algorithm of [20], and in time $O(s\log^c p)$ for some constant $c$, using the method of [21]. Though beyond the scope of this paper, a detailed computational study of the incorporation of these various linear system solvers into our approach, as well as the designing of linear systems solvers targeted to the specific structure underlying our closed-form estimators, are certainly very interesting directions for future work.

## C   Useful lemma(s)

**Lemma 1** (Theorem 1 of [15, 17]). *Let $\delta$ be $\max_{ij}\left|\left[\frac{X^\top X}{n}\right]_{ij} - \Sigma_{ij}\right|$. Suppose that $\nu \geq 2\delta$. Then, under the conditions (C3) and the properties of a generalized thresholding operator described in Section 3, we can deterministically guarantee that the spectral norm of error is bounded as follows*

$$\left\|T_\nu\left(\frac{X^\top X}{n}\right) - \Sigma\right\|_\infty \leq 5\nu^{1-q}c_0 + 3\nu^{-q}c_0\delta. \tag{10}$$

**Lemma 2** (Lemma 1 of [22]). *Let $\mathcal{A}$ be the event that*

$$\left\|\frac{X^\top X}{n} - \Sigma\right\|_\infty \leq 8(\max_i \Sigma_{ii})\sqrt{\frac{10\tau \log p'}{n}}$$

*where $p' := \max\{n, p\}$ and $\tau$ is any constant greater than 2. Suppose that the design matrix $X$ is i.i.d. sampled from $\Sigma$-Gaussian ensemble with $n \geq 40\max_i \Sigma_{ii}$. Then, the probability of event $\mathcal{A}$ occurring is at least $1 - 4/p'^{\tau-2}$.*

## D   Proof of Theorem 1

In order to prove the theorem, we first generalize theorems in [14, 23, 24] for GLMs:

**Theorem 2.** *Suppose we solve the estimation problem (4), such that true structured moment satisfies Condition (C1), and the constraint term $\lambda_n$ is set as $\lambda_n \geq \left\|\theta^* - \left[T_\nu\left(\frac{X^\top X}{n}\right)\right]^{-1}\frac{X^\top[\nabla A]^{-1}(\Pi_{\bar{\mathcal{M}}}(y))}{n}\right\|_\infty$. Then, the optimal solution $\widehat{\theta}$ of (4) satisfies:*

$$\|\widehat{\theta} - \theta^*\|_\infty \leq 2\lambda_n,$$
$$\|\widehat{\theta} - \theta^*\|_2 \leq 4\sqrt{k}\lambda_n,$$
$$\|\widehat{\theta} - \theta^*\|_1 \leq 8k\lambda_n.$$

To complete the proof, we need to show that $\lambda_n \geq \left\|\theta^* - \left[T_\nu\left(\frac{X^\top X}{n}\right)\right]^{-1}\frac{X^\top[\nabla A]^{-1}(\Pi_{\bar{\mathcal{M}}}(y))}{n}\right\|_\infty$ under the conditions specified in Theorem 1.

We first compute the upper bound of $\left\|\left[T_\nu\left(\frac{X^\top X}{n}\right)\right]^{-1}\right\|_\infty$, so that we later have

$$\left\|\theta^* - \left[T_\nu\left(\frac{X^\top X}{n}\right)\right]^{-1}\frac{X^\top[\nabla A]^{-1}(\Pi_{\bar{\mathcal{M}}}(y))}{n}\right\|_\infty$$
$$\leq \left\|\left[T_\nu\left(\frac{X^\top X}{n}\right)\right]^{-1}\right\|_\infty\left\|T_\nu\left(\frac{X^\top X}{n}\right)\theta^* - \frac{X^\top[\nabla A]^{-1}(\Pi_{\bar{\mathcal{M}}}(y))}{n}\right\|_\infty$$
$$\leq \left\|\left[T_\nu\left(\frac{X^\top X}{n}\right)\right]^{-1}\right\|_\infty\left(\left\|\frac{X^\top}{n}\left\{X\theta^* - [\nabla A]^{-1}(\Pi_{\bar{\mathcal{M}}}(y))\right\}\right\|_\infty + C_1\sqrt{\frac{\log p'}{n}}\|\theta^*\|_1\right)$$

where we use the fact that $\|T_\nu\left(\frac{X^\top X}{n}\right)\theta^* - \frac{X^\top X}{n}\theta^*\|_\infty \leq C_1\sqrt{\frac{\log p'}{n}}\|\theta^*\|_1$ given the selection of $\nu$ in the statement.

Furthermore, with the selection $\nu$ in the statement, Lemma 1 and 2 hold with probability at least $1 - 4/p'^{\tau-2}$. Armed with (10), we use the triangle inequality of norm and the condition (C3): for any $w$

$$\left\|T_\nu\left(\frac{X^\top X}{n}\right)w\right\|_\infty = \left\|T_\nu\left(\frac{X^\top X}{n}\right)w - \Sigma w + \Sigma w\right\|_\infty \geq \|\Sigma w\|_\infty - \left\|\left(T_\nu\left(\frac{X^\top X}{n}\right) - \Sigma\right)w\right\|_\infty$$

$$\overset{(i)}{\geq} \kappa_\ell\|w\|_\infty - \left\|\left(T_\nu\left(\frac{X^\top X}{n}\right) - \Sigma\right)w\right\|_\infty \geq \left(\kappa_\ell - \left\|T_\nu\left(\frac{X^\top X}{n}\right) - \Sigma\right\|_\infty\right)\|w\|_\infty$$

where the inequality (i) uses the condition (C3). Now, by Lemma 1 with the selection of $\nu$, we have

$$\left\|T_\nu\left(\frac{X^\top X}{n}\right) - \Sigma\right\|_\infty \leq c_1\left(\frac{\log p'}{n}\right)^{(1-q)/2} c_0$$

where $c_1$ is a constant related only on $\tau$ and $\max_i \Sigma_{ii}$. Specifically, it is defined as $6.5\left(16(\max_i \Sigma_{ii})\sqrt{10\tau}\right)^{1-q}$. Hence, as long as $n > \left(\frac{2c_1c_0}{\kappa_\ell}\right)^{\frac{2}{1-q}}\log p'$ as stated, so that $\left\|T_\nu\left(\frac{X^\top X}{n}\right) - \Sigma\right\|_\infty \leq \frac{\kappa_\ell}{2}$, we can conclude that $\left\|T_\nu\left(\frac{X^\top X}{n}\right)w\right\|_\infty \geq \frac{\kappa_\ell}{2}\|w\|_\infty$, which implies $\left\|\left[T_\nu\left(\frac{X^\top X}{n}\right)\right]^{-1}\right\|_\infty \leq \frac{2}{\kappa_\ell}$

Therefore, now we have

$$\left\|\theta^* - \left[T_\nu\left(\frac{X^\top X}{n}\right)\right]^{-1}\frac{X^\top[\nabla A]^{-1}\left(\Pi_{\bar{\mathcal{M}}}(y)\right)}{n}\right\|_\infty$$

$$\leq \frac{2}{\kappa_\ell}\left(\left\|\frac{X^\top}{n}\left\{X\theta^* - [\nabla A]^{-1}\left(\Pi_{\bar{\mathcal{M}}}(y)\right)\right\}\right\|_\infty + C_1\sqrt{\frac{\log p'}{n}}\|\theta^*\|_1\right). \tag{11}$$

To finalize the proof, we now focus on the term $\left\|\frac{X^\top}{n}\left\{X\theta^* - [\nabla A]^{-1}\left(\Pi_{\bar{\mathcal{M}}}(y)\right)\right\}\right\|_\infty$ in (11). Noting that $\nabla A$ and $[\nabla A]^{-1}$ are element-wise functions, (11) can be rewritten as

$$\left\|\frac{X^\top}{n}\left\{[\nabla A]^{-1}\left(\nabla A(X\theta^*)\right) - [\nabla A]^{-1}\left(\Pi_{\bar{\mathcal{M}}}(y)\right)\right\}\right\|_\infty. \tag{12}$$

By applying mean value theorem to every element in $[\nabla A]^{-1}\left(\nabla A(X\theta^*)\right) - [\nabla A]^{-1}\left(\Pi_{\bar{\mathcal{M}}}(y)\right)$, we have, for a fixed $j \in \{1,2,\ldots,p\}$,

$$\left|\left[\frac{X^\top}{n}\left\{[\nabla A]^{-1}\left(\nabla A(X\theta^*)\right) - [\nabla A]^{-1}\left(\Pi_{\bar{\mathcal{M}}}(y)\right)\right\}\right]_j\right|$$

$$= \left|\frac{1}{n}\sum_{i=1}^n X_{ij}\left\{(A')^{-1}\left(A'(\langle\theta^*, x^{(i)}\rangle)\right) - (A')^{-1}\left(\left[\Pi_{\bar{\mathcal{M}}}(y)\right]_i\right)\right\}\right|$$

$$= \left|\frac{1}{n}\sum_{i=1}^n X_{ij}L_i\left\{A'(\langle\theta^*, x^{(i)}\rangle) - \left[\Pi_{\bar{\mathcal{M}}}(y)\right]_i\right\}\right|$$

where $L_i$ is $[(A')^{-1}]'(a)$ for some point $a$ between $A'(\langle\theta^*, x^{(i)}\rangle)$ and $\left[\Pi_{\bar{\mathcal{M}}}(y)\right]_i$ by mean value theorem. By triangular inequality of $\ell_\infty$ norm, we have

$$\left|\frac{1}{n}\sum_{i=1}^n X_{ij}L_i\left\{A'(\langle\theta^*, x^{(i)}\rangle) - y^{(i)} + y^{(i)} - \left[\Pi_{\bar{\mathcal{M}}}(y)\right]_i\right\}\right|$$

$$\leq \underbrace{\left|\frac{1}{n}\sum_{i=1}^n X_{ij}L_i\left\{A'(\langle\theta^*, x^{(i)}\rangle) - y^{(i)}\right\}\right|}_{(I)} + \underbrace{\left|\frac{1}{n}\sum_{i=1}^n X_{ij}L_i\left\{y^{(i)} - \left[\Pi_{\bar{\mathcal{M}}}(y)\right]_i\right\}\right|}_{(II)}. \tag{13}$$

(*Upper bound of (I) in* (13)): (I) can be upper bounded as shown in [8]:

$$\max_{j=1,\ldots,p}\left|\frac{1}{n}\sum_{i=1}^{n}X_{ij}L_i\big\{A'(\langle\theta^*,x^{(i)}\rangle)-y^{(i)}\big\}\right|\le\kappa_u\max_i|L_i|\sqrt{2\max_i|A''(\langle\theta^*,x^{(i)}\rangle)|\frac{\log p'}{n}}$$

$$\tag{14}$$

with probability at least $1-cp'^{-c'}$. By condition (C2), we have

$$\mathbb{P}(|\langle\theta^*,x^{(i)}\rangle|\ge\delta)\le 2\exp\left(-\frac{\delta^2}{2\kappa_u^2\|\theta^*\|_2^2}\right)\quad\text{for all }\delta>0,\tag{15}$$

hence it follows $\max_i|\langle\theta^*,x^{(i)}\rangle|\le 2\|\theta^*\|_2\kappa_u\sqrt{\log n}$ with probability at least $1-2/n$. Now, since for all $a$ in $\mathcal{M}'\cup\overline{\mathcal{M}}$, $|[(A')^{-1}]'(a)|\le\kappa_{\ell,A}$, and $\max_{i=1,\ldots,n}|A''(\langle\theta^*,x^{(i)}\rangle)|\le\kappa_{u,A}$ by definition in (8), we simply obtain

$$\text{(I)}\le\kappa_u\kappa_{\ell,A}\sqrt{2\kappa_{u,A}\frac{\log p'}{n}}\,.\tag{16}$$

(*Upper bound of (II) in* (13)): For a fixed $j\in\{1,2,\ldots,p\}$, the variable $\{X_{ij}\}_{i=1}^n$ are i.i.d., zero mean and sub-Gaussian by the condition (C2). Hence with vector $\left[L_i(y^{(i)}-[\Pi_{\overline{\mathcal{M}}}(y)]_i)\right]_{i=1}^n$ whose $\ell_2$ norm is bounded by $\sqrt{n}\max_{i=1,\ldots,n}L_i\big(y^{(i)}-\big[\Pi_{\overline{\mathcal{M}}}(y)\big]_i\big)$, we have

$$\max_{j=1,\ldots,p}\left|\frac{1}{n}\sum_{i=1}^{n}X_{ij}L_i\big\{y^{(i)}-\big[\Pi_{\overline{\mathcal{M}}}(y)\big]_i\big\}\right|\le 2\kappa_u\left(\max_{i=1,\ldots,n}L_i\big(y^{(i)}-\big[\Pi_{\overline{\mathcal{M}}}(y)\big]_i\big)\right)\sqrt{\frac{\log p'}{n}}$$

with probability at least $1-cp'^{-c'}$. Combining all pieces together completes the proof.

The proof of the statement (b) is straightforward. As we have just shown, $\Big\|\theta^*-\big[T_\nu\big(\frac{X^\top X}{n}\big)\big]^{-1}\frac{X^\top[\nabla A]^{-1}(\Pi_{\overline{\mathcal{M}}}(y))}{n}\Big\|_\infty\le\lambda_n$. Moreover, our estimator $\widehat{\theta}$ is obtained by applying element-wise soft-thresholding from $\big[T_\nu\big(\frac{X^\top X}{n}\big)\big]^{-1}\frac{X^\top[\nabla A]^{-1}(\Pi_{\overline{\mathcal{M}}}(y))}{n}$. Hence, if $\theta_i^*=0$, then $\Big|\big[\big[T_\nu\big(\frac{X^\top X}{n}\big)\big]^{-1}\frac{X^\top[\nabla A]^{-1}(\Pi_{\overline{\mathcal{M}}}(y))}{n}\big]_i\Big|\le\lambda_n$ and $\widehat{\theta}_i=0$ after thresholding by construction. At the same time, if $|\theta_i^*|$ is greater than $3\lambda_n$, it will be trivially non-zero even after thresholding since $\|\widehat{\theta}-\theta^*\|_\infty\le 2\lambda_n$, therefore $\widehat{\theta}$ correctly includes all such true supports.

## E   Applying Theorem 1 for exponential regression models

Exponential regression models described in Section 2 require the covariates are necessarily bounded so that the canonical parameter should be strictly negative and the distribution can be normalizable. Hence in this case, we will assume the parameter has a bias term $b<0$, so that the canonical parameter takes the form $\langle\theta^*,x\rangle+b$. To make it sure every canonical parameter is strictly negative, $x^{(i)}$ is properly bounded and/or the bias $b$ is smaller enough. Toward this, we assume $b\le-2\eta_{\max}$ so that $\langle\theta^*,x^{(i)}\rangle+b\le-\eta_{\max}<0$ for all $x^{(i)}$.

Now, we specify $\kappa_{\ell,A}$ and $\kappa_{u,A}$ as in other cases. Since the exponential family log-partition function for exponential regression models is $A(\eta):=-\log(-\eta)$, we have $|A''(\langle\theta^*,x^{(i)}\rangle+b)|=\frac{1}{(\langle\theta^*,x^{(i)}\rangle+b)^2}\le\frac{1}{\eta_{\max}^2}$ for all $i=1,\ldots,n$, so that (8) holds with $\kappa_{u,A}=\frac{1}{\eta_{\max}^2}$.

We now turn to the inequality on $\kappa_{\ell,A}$ in (8). In this case, the response moment polytope for the response $y\in\mathcal{Y}\equiv[0,\infty)$ is given by $\mathcal{M}=[0,\infty)$. Its interior in turn is given by $\mathcal{M}^o=(0,\infty)$. For our subset of the interior we thus set $\overline{\mathcal{M}}=[\epsilon,\infty)$, for some $\epsilon$ s.t. $0<\epsilon<1$. The forward mapping of exponential models is $A'(\eta)=-\frac{1}{\eta}$, and therefore $\mathcal{M}'$ in (9) becomes $[\frac{1}{\eta_{\max}},\infty)$. Then, noting that the inverse mapping of exponential regression models is given by $(A')^{-1}(\mu)=-\frac{1}{\mu}$, it is Lipschitz for $\overline{\mathcal{M}}\cup\mathcal{M}'$ with constant $\kappa_{\ell,A}\max\{\frac{1}{\eta_{\max}^2},1/\epsilon^2\}$, so that the inequality on $\kappa_{\ell,A}$ in (8) holds as well.

Note that with this setting of $\overline{\mathcal{M}}$, $\Pi_{\overline{\mathcal{M}}}(y_i)=\mathcal{I}(y_i\le\epsilon)\epsilon+\mathcal{I}(y_i>\epsilon)y_i$, where $\mathcal{I}(A)$ is an indicator function that is equal to $1$ if $A$ is true and $0$ otherwise. Then, we can directly derive the results as in Corollary 2

Figure 1: Elem-GLM vs. $\ell_1$ regularized MLE for logistic regression models when $(n, p, k) = (1000, 1000, 10)$ (Left) and $(n, p, k) = (1000, 5000, 10)$ (Right).

Figure 2: Elem-GLM vs. $\ell_1$ regularized MLE for Poisson regression models when $(n, p, k) = (300, 200, 10)$ (Left) and $(n, p, k) = (300, 2000, 10)$ (Right).

## F    Experiment Details and Additional Experiments

We find $c, c'$ from the set with a wide range: $C := \{10^2, 75, 50, 25, 10, \dots, 10^{-9}, 7.5 \times 10^{-10}, 5 \times 10^{-10}, 2.5 \times 10^{-10}, 10^{-10}\}$. We compare our estimator with $\ell_1$ regularized MLEs with different stopping criteria: (i) $\ell_1$ MLE[1] stops the descent algorithm after the number of iterations exceeds $i_{\max} = 50$, (ii) $\ell_1$ MLE[2] stops after $i_{\max} = 10^2$, and (iii) $\ell_1$ MLE[3] does after $i_{\max} = 500$ iterations. Or they can stop if no update is made at certain iteration $i$ before $i_{\max}$: $\frac{\|\theta^i - \theta^{i-1}\|_2}{\|\theta^{i-1}\|_2}$ is smaller than $\delta = 10^{-10}$.

For fair comparisons, we again find the regularization parameter of regularized MLEs, $\lambda_n = c\sqrt{\log p / n}$ where $c \in C$ from a validation set. Moreover, we apply an element-wise soft-thresholding even for regularized MLEs with thresholding parameter $c'\sqrt{\log p / n}$ where $c' \in C$. Experiments here are run on MATLAB in a single computing node with a Intel Core i5 2.5GHz CPU and 8G memory. We solve $\ell_1$ regularized MLE using proximal gradient descent, and employ the built-in MATLAB function "\" to solve linear system involving $T_\nu\left(\frac{X^\top X}{n}\right)$.

Figure 1 shows receiver operator curves (ROC) for support set recovery task of logistic regression models under two different regimes when varying regularization parameters $\lambda_n$ for both methods. The thresholding parameter for Elem-GLM is selected via cross validation as described above. Similarly, Figure 2 represents the results for Poisson regression models. For Poisson model, we randomly choose non-zero $k$ entries in $\theta^*$ from uniform distribution in $(1, 2)$.

Table 2-4 show the results of experiments described in Section 5, some of which are preselected and provided in Table 1.

Table 2: Comparisons on simulated datasets when parameters are tuned to minimize $\ell_2$ error on independent validation sets. We fix $p = 2000$ and vary $n$ and $k$.

| $(n, k)$ | METHOD | TP | FP | $\ell_2$ ERROR | TIME (SEC) |
|---|---|---|---|---|---|
| (1000,10) | $\ell_1$ MLE[1] | 1 | 0.1410 | 4.6558 | 11.9 |
| | $\ell_1$ MLE[2] | 1 | 0.1407 | 4.2583 | 23.2 |
| | $\ell_1$ MLE[3] | 1 | 0.1609 | 3.6589 | 87.9 |
| | **ELEM-GLM** | 1 | 0.0156 | **3.0954** | **2.9** |
| (1000,100) | $\ell_1$ MLE[1] | 0.9240 | 0.2771 | 19.4423 | 12.4 |
| | $\ell_1$ MLE[2] | 0.9330 | 0.2895 | 19.2349 | 24.6 |
| | $\ell_1$ MLE[3] | 0.9340 | 0.3304 | 19.0301 | 90.6 |
| | **ELEM-GLM** | 0.7890 | 0.1918 | **16.7169** | **2.9** |
| (1500,10) | $\ell_1$ MLE[1] | 1 | 0.1719 | 4.4271 | 16.9 |
| | $\ell_1$ MLE[2] | 1 | 0.1751 | 3.9717 | 33.6 |
| | $\ell_1$ MLE[3] | 1 | 0.1685 | 3.2796 | 124.1 |
| | **ELEM-GLM** | 1 | 0.0606 | **2.9346** | **3.7** |
| (1500,100) | $\ell_1$ MLE[1] | 0.9820 | 0.3153 | 19.0947 | 18.2 |
| | $\ell_1$ MLE[2] | 0.9800 | 0.3465 | 18.8142 | 36.0 |
| | $\ell_1$ MLE[3] | 0.9820 | 0.4023 | 18.5004 | 129.6 |
| | **ELEM-GLM** | 0.8700 | 0.1636 | **15.2261** | **3.8** |

We also evaluate the performance of our closed-form estimators on some real binary classification datasets, obtained from LIBSVM (http://www.csie.ntu.edu.tw/~cjlin/libsvmtools/datasets/). We assume the logistic regression models for these datasets, and compare our Elem-GLM against $\ell_1$ MLE[1] with $i_{\max} = 10^2$ and $\ell_1$ MLE[2] with $i_{\max} = 10^3$. Since the true parameter $\theta^*$ is unknown for real datasets, we tune parameters for a classification error rates on a new test set and evaluate the estimators in terms of classification error rates on a test set and entire training time (in second) for cross-validation using the set $C$ above. Toward this, we divide each dataset into equal-sized 9 parts and combine 3 parts to generate training, validation and test sets; we combine them in 9 different ways and average the results over 9 different cases. Results are summarized in Table 5. Note that datasets here are not high-volume and some are not even high-dimensional $(p < n)$. We defer comparing estimators on big-data sets to future work.

Table 3: Comparisons on simulated datasets when parameters are tuned to minimize $\ell_2$ error on independent validation sets. We fix $p = 5000$ and vary $n$ and $k$.

| $(n, k)$ | METHOD | TP | FP | $\ell_2$ ERROR | TIME (SEC) |
|---|---|---|---|---|---|
| (2000,10) | $\ell_1$ MLE[1] | 1 | 0.1094 | 4.5450 | 63.9 |
| | $\ell_1$ MLE[2] | 1 | 0.0873 | 4.0721 | 133.1 |
| | $\ell_1$ MLE[3] | 1 | 0.1000 | 3.4846 | 348.3 |
| | **ELEM-GLM** | 0.9900 | 0.0184 | **2.7375** | **26.5** |
| (2000,100) | $\ell_1$ MLE[1] | 0.9910 | 0.1595 | 19.2499 | 71.5 |
| | $\ell_1$ MLE[2] | 0.9900 | 0.1658 | 18.9967 | 141.8 |
| | $\ell_1$ MLE[3] | 0.9900 | 0.1887 | 18.7371 | 355.1 |
| | **ELEM-GLM** | 0.9180 | 0.1517 | **15.5974** | **26.2** |
| (2000,1000) | $\ell_1$ MLE[1] | 0.7004 | 1 | 65.3568 | 72.0 |
| | $\ell_1$ MLE[2] | 0.6974 | 1 | 65.3106 | 141.3 |
| | $\ell_1$ MLE[3] | 0.6949 | 1 | 65.2976 | 388.9 |
| | **ELEM-GLM** | 0.7091 | 0.9541 | **63.4709** | **25.9** |
| (4000,10) | $\ell_1$ MLE[1] | 1 | 0.1626 | 4.2132 | 155.5 |
| | $\ell_1$ MLE[2] | 1 | 0.1327 | 3.6569 | 296.8 |
| | $\ell_1$ MLE[3] | 1 | 0.1112 | 2.9681 | 829.3 |
| | **ELEM-GLM** | 1 | 0.0069 | **2.6213** | **40.2** |
| (4000,100) | $\ell_1$ MLE[1] | 1 | 0.2208 | 18.7428 | 166.5 |
| | $\ell_1$ MLE[2] | 1 | 0.2345 | 18.3571 | 318.0 |
| | $\ell_1$ MLE[3] | 1 | 0.2962 | 17.9892 | 838.6 |
| | **ELEM-GLM** | 0.9950 | 0.4381 | **15.9836** | **40.4** |
| (4000,1000) | $\ell_1$ MLE[1] | 0.7086 | 1 | 65.1418 | 159.9 |
| | $\ell_1$ MLE[2] | 0.7021 | 1 | 65.0396 | 320.9 |
| | $\ell_1$ MLE[3] | 0.6999 | 0.9999 | 64.9398 | 950.6 |
| | **ELEM-GLM** | 0.8007 | 1 | **62.8094** | **40.6** |

Table 4: Comparisons on simulated datasets when parameters are tuned to minimize $\ell_2$ error on independent validation sets. We fix $p = 10^4$ and vary $n$ and $k$.

| $(n, k)$ | METHOD | TP | FP | $\ell_2$ ERROR | TIME (SEC) |
|---|---|---|---|---|---|
| (5000,100) | $\ell_1$ MLE[1] | 1 | 0.1301 | 18.9079 | 500.1 |
| | $\ell_1$ MLE[2] | 1 | 0.1695 | 18.5567 | 983.8 |
| | $\ell_1$ MLE[3] | 1 | 0.2001 | 18.2351 | 2353.3 |
| | **ELEM-GLM** | 0.9975 | 0.3622 | **16.4148** | **151.8** |
| (5000,1000) | $\ell_1$ MLE[1] | 0.7990 | 1 | 65.1895 | 520.7 |
| | $\ell_1$ MLE[2] | 0.7935 | 1 | 65.1165 | 1005.8 |
| | $\ell_1$ MLE[3] | 0.7965 | 1 | 65.1024 | 2560.1 |
| | **ELEM-GLM** | 0.8295 | 1 | **63.2359** | **152.1** |
| (8000,100) | $\ell_1$ MLE[1] | 1 | 0.1904 | 18.6186 | 810.6 |
| | $\ell_1$ MLE[2] | 1 | 0.2181 | 18.1806 | 1586.2 |
| | $\ell_1$ MLE[3] | 1 | 0.2364 | 17.6762 | 3568.9 |
| | **ELEM-GLM** | 0.9450 | 0.0359 | **11.9881** | **221.1** |
| (8000,1000) | $\ell_1$ MLE[1] | 0.7965 | 1 | 65.0714 | 809.5 |
| | $\ell_1$ MLE[2] | 0.7900 | 1 | 64.9650 | 1652.8 |
| | $\ell_1$ MLE[3] | 0.7865 | 1 | 64.8857 | 4196.6 |
| | **ELEM-GLM** | 0.7015 | 0.5103 | **61.0532** | **219.4** |

Table 5: Comparisons of the empirical prediction errors on some benchmark datasets.

| DATA SET | $\ell_1$ MLE[1] | | $\ell_1$ MLE[2] | | **Elem-GLM** | |
|---|---|---|---|---|---|---|
| | ERR | TIME | ERR | TIME | ERR | TIME |
| COD-RNA | 0.3156 | 3.61 | 0.0669 | 133.9 | **0.0640** | **0.25** |
| IJCNN1 | 0.0971 | 5.10 | **0.0771** | 164.5 | 0.0842 | **0.25** |
| DIABETES | 0.2596 | 0.09 | 0.2356 | 1.09 | **0.2313** | **0.01** |
| HEART | **0.1938** | 0.06 | 0.2062 | 1.55 | 0.1975 | **0.01** |
| AUSTRALIAN | 0.1448 | 0.07 | 0.1352 | 4.68 | **0.1347** | **0.01** |