[Reviews · NeurIPS 2015]

Submitted by Assigned_Reviewer_1

In this paper, the authors propose closed-form estimators for GLMs in high-dimensional setting. Since I am not so familiar with this field, it seems difficult for me to evaluate the novelty and the correctness of this paper. However, the paper looks comprehensive with sufficient theoretical evidences and relevant to ML. Here, I just pose a few minor question/concerns to the authors.

First, I wonder how we can specify several quantities or a subset \overline{\mathcal{M}} for specific instances of GLMs. In this point, the authors kindly provide the examples in Sec. 4 (logistic regression and poisson regression), which are in fact useful. However, it would be more useful if the author give some guideline or intuitive explanations to set up these. Especially, as for \overline{\mathcal{M}}, although the authors state it is ``carefully" chosen, this description seems a bit ambiguous and may mystify readers.

In Remark 1, the authors mention their framework can be applied to the cases of structured regularizations, such as group sparsity. This would be very interesting and one of the key advantages of the framework from the viewpoint of generality. However again, it would be ambiguous how the quantities can be setup for each specific structured regularizer.

The empirical results seem to show in some degree that the proposed estimator works reasonably well as expected from the theoretical results.
Summary: In this paper, the authors propose closed-form estimators for GLMs in high-dimensional setting. Since I am not so familiar with this field, it seems difficult for me to evaluate the novelty and the correctness of this paper. However, the paper looks comprehensive with sufficient theoretical evidences and relevant to ML.

Submitted by Assigned_Reviewer_2

This paper introduces elem-glm, a novel estimator for the parameter in a generalized linear model in high dimension. The proposed estimator is based on an approximation of the maximum likelihood estimator for which a closed form is available. This approximation is further thresholded for regularization purpose. Consistency and sparsistency results are proposed for the resulting estimator.

I found this paper refreshing. This approach is very original and orthogonal to most recent work on estimators in high dimension which either introduce new regularizers or faster optimization algorithms. By approximating the MLE in a smart way, the authors obtain a novel estimator which is faster and easier to compute, yet enjoys good statistical properties. I think this is likely to inspire other researchers and create more diversity in future papers on high dimensional statistics. The paper is also rigorous, remarkably clear and well written.

A usual workaround when computing sparse estimators over large p is to use active set techniques. Both elem-glm (using formulation (6)) and the iterative proximal algorithm against which it is compared could benefit from using an active set for large p, but it would be interesting to quantify if the closed form still leads to a large computational gain in this context.

The non-invertibility of the sample covariance is dealt with using thresholding. I understand this is useful in the derivation of Theorem 1, but would it be possible to obtain a similar result (under a different C3) using ridge regularization, or is there something specific to thresholding here?

Finally, the subgaussian design is a rather strong assumption and one that is not always made when studying consistency of GLM estimators. It would be interesting to show how elem-glm performs when this assumption is violated, eg when X is binary.

Minor points:

- l. 170-171 repeats some notation that was introduced in 2.2. - l. 216: a word seems to be missing: "A'(.) is only onto the interior

of M" -> "invertible onto"?
Summary: An excellent paper from every point of view (quality, clarity, originality and significance).

Submitted by Assigned_Reviewer_3

- Summary of Paper

- The paper approaches the problem of estimating parameters in GLMs

when the dimension of the data is larger than the number of

samples. The authors claim to have a closed-form solution for

such a setting based on thresholding and projection operations.

The closed-form estimators enjoy some favorable frequentist

statistical properties.

Finally, the authors compare their

estimates to l1 regularized MLE estimates for two simulated data

sets. - Quality

- I have questions (L120 comments below) about the work that trace

back to the way the authors formulated the exponential family

form of logistic regression and poisson regression. This issue

impacts how the authors define the subset of the interior of the

mean-response polytope and therefore the evidence provided in the

results section.

- L120: The authors' definition of the logistic regression is

confusing and I'm not sure if it is correct. I might write the

exponential family as $p(y_i | theta, x) = exp( y__i * \eta -

A(\eta)$, where $\eta = log(\mu/(1-\mu))$ is the log-odds, the

log-partition function is $A(\eta) = log(1+\exp(\eta)$, and $\mu

= (1+\exp(\theta*x))^{-1}$ is the logistic function. This

formulation is very different than the one presented: $\exp{y*z -

\log(\exp{z}+\exp{-z}}$ where z = < \theta, x > is their natural

parameter. In particular the stated log partition function is

A(a) = log(exp(-a) + exp(a)). We could write that as A(a) =

log(exp(-a^2)). How did the authors arrive at the stated form for

the logistic regression exponential family density function?

- L120: I am similarly confused by how the authors arrived at their

exponential family formulation for the poisson regression

density. How did the authors arrive at this formulation and can

they show that it is certainly correct?

- L296: The theorem seems to provide for not missing an important

predictor with high probability, but we would also like to make

sure to set coefficient to zero that should be. Is this theorem

one-sided in that it only provides for not missing a non-zero

coefficient?

- L237: The article develops reasons for truncating (X^T * X) and

for the projection on the subset of the polytope in the

estimator. But the reason for the soft-threshold

S_{\lambda_n}

is not clear. This should be clarified.

- L296: The theorem shows that the estimator includes all non-zero

supports with a probability bound. Is this bound non-trivial in

any specific instances? - Clarity

- The paper is fairly clear, though dense. A few edits outlined

below could make it more clear.

- L041: The introductory paragraph starts with talking about

estimation and ends with talking about prediction. While these

concepts are related, it would help to be consisent so early in

the manuscript.

- L046: References would help after "and others"

- L106: The notation for the natural parameter in the exponential

family model < \theta, x > is non-standard and confusing. The

bracket notation is used by some to denote dot-product and other

to denote expected value. A more familar notation or a

justification for this particular notation would help the article

clarity.

- L118: The use of "A(a)" is confusing and "a" could be changed for

the authors' notation for the natural parameter < \theta, x >.

- L190: The article seems to rely heavily on \hat{M} the subset of

the interior of the response moment polytope. But we don't get to

see what that is until we get to specific models. It would help

to remind the reader that \hat{M} will be defined for specific

models more frequently.

- L194: The estimator is stated without development. Either put the

estimator after the development or state that development will

follow. The "machinery" is not described, it is the notation that

is described earlier. The "machinery" seems to be described

later. The reason for "Elem" subscript on \theta is not described.

- L322: Remark 3 refers to "k". I could not find what "k" is

earlier in the paper. - Originality

- The paper builds on some previous work on linear regression, but

the extension to the GLM setting is novel. - Significance

- As the authors state, even the MLE in a low-dimensional setting p

< n does not have a closed-form solution. So, the development of

a GLM estimator that is both closed form and valid in high

dimensional settings is highly significant.
Summary: The paper proposes a significant advance in inference for a broad class of statistical models in a high dimensional setting. If several technical issues are clarified the article could be an important development.

Author Feedback
Author rebuttal: We thank the reviewers for the kind comments on the novelty and potential impact of the paper.

Assigned_Reviewer_1:

As shown in the examples of logistic and Poisson regressions, the selection of \overline{\mathcal{M}} plays a key role in specifying the several quantities in Theorem 1. Here there are two main issues to consider: (a) \overline{\mathcal{M}} should be small so that \kappa_{\ell,A} in eq. (8) is properly bounded
while, (b) as an approximation, \overline{\mathcal{M}} should be close enough to \mathcal{M}. This is quantified in \Epsilon defined in Theorem 1. Given a specific instance of GLM, it should be carefully selected in between these two extremes to utilize Theorem 1 and have meaningful bounds.

For general cases of structured parameter estimation, the target regularizer (for instance, group \ell_1 norm for group sparsity case) and its dual norm (group \ell_\infty) can be used in the objective and the constraint, respectively, rather than \ell_1 and \ell_\infty norms in eq. (6).

We will add discussion expanding upon the above to the final version.

Assigned_Reviewer_2:

It would be very interesting and an important future research direction to consider and find other families of covariates and the corresponding plug-in covariance estimators, for handling non-invertible sample covariance. The proposed method under (C3) is one example where we can provide strong statistical guarantees.

Sub-Gaussian assumption is rather mild (Note that it holds even for bounded e.g. the binary values asked by the reviewer); nevertheless the study for cases where some conditions are violated would be interesting future work.

Assigned_Reviewer_3:
- L120: Logistic regression can be re-parameterized in several ways, and please note that the domains of the response are different in our formulation (where it is {-1,1}) and what you are probably considering (where it is {0,1}). The log-partition function \log( \exp(-\eta)+\exp(\eta) ) in our formulation will be \log( 1+ \exp(\eta) ) when the domain is {0,1} since \eta = 0 implies \exp(-\eta) is 1.
- L120: Expression of Poisson distribution: one can arrive at the Poisson distribution as an exponential family by simply substituting \lambda = \exp(\eta) from the standard formulation of Poisson distribution.
- L296: Our statement says that our method can *exclude* (or set to zero) all true zero values (this is one line above, in line 295), guaranteeing sparsistency.
- L322: 'k' is defined in (C1)

Assigned_Reviewer_4 and Assigned_Reviewer_5:

We require a slightly more stringent condition in theory than \ell_1 MLE estimators for estimation error guarantees: that the covariance matrix of covariates X satisfy (C3) under which our closed-form expression allows for efficient computation.

On tuning parameters: once we have initial estimator for a specific selection of \nu, computing the entire path of \lambda_n is trivial: element-wise soft-thresholding with different amounts. Moreover, we can also see that the selection of \epsilon is not sensitive in Theorem 1 because that will only affect the constants: kappa_{\ell,A} in (8) and | y^(i) - [\proj(y)]_i | in \Epsilon.

Assigned_Reviewer_6:

Instead of inverting huge p x p matrix, our estimator in (4) can be obtained by solving the linear system involving the thresholded sample covariance matrix --- which is *sparse* by construction. For such sparse linear systems, there exist very efficient methods - many recently proposed over the last decade. For example, consider the case where the matrix is diagonally dominant then inverting such matrix can be done in quasi-linear time w.r.t to the amount of sparsity. [Spielman&Teng 2003].